# Crucial Role of Oncogenic *KRAS* Mutations in Apoptosis and Autophagy Regulation: Therapeutic Implications

**DOI:** 10.3390/cells11142183

**Published:** 2022-07-13

**Authors:** Anabela Ferreira, Flávia Pereira, Celso Reis, Maria José Oliveira, Maria João Sousa, Ana Preto

**Affiliations:** 1Centre of Molecular and Environmental Biology (CBMA), Department of Biology, Campus de Gualtar, University of Minho, 4710-057 Braga, Portugal; aaferreira18@gmail.com (A.F.); flaviabrandao.fcbp@gmail.com (F.P.); mjsousa@bio.uminho.pt (M.J.S.); 2Institute of Science and Innovation for Bio-Sustainability (IB-S), Campus de Gualtar, University of Minho, 4710-057 Braga, Portugal; 3Institute for Research and Innovation in Health (i3S), University of Porto, 4200-135 Porto, Portugal; celsor@ipatimup.pt (C.R.); mariajo@ineb.up.pt (M.J.O.); 4Institute of Biomedical Engineering (INEB), University of Porto, 4200-135 Porto, Portugal; 5Institute of Biomedical Sciences Abel Salazar (ICBAS), University of Porto, 4050-313 Porto, Portugal; 6Institute of Molecular Pathology and Immunology of the University of Porto (IPATIMUP), 4200-135 Porto, Portugal

**Keywords:** *KRAS* mutations, cell death resistance, apoptosis, autophagy

## Abstract

KRAS, one of the RAS protein family members, plays an important role in autophagy and apoptosis, through the regulation of several downstream effectors. In cancer cells, *KRAS* mutations confer the constitutive activation of this oncogene, stimulating cell proliferation, inducing autophagy, suppressing apoptosis, altering cell metabolism, changing cell motility and invasion and modulating the tumor microenvironment. In order to inhibit apoptosis, these oncogenic mutations were reported to upregulate anti-apoptotic proteins, including Bcl-xL and survivin, and to downregulate proteins related to apoptosis induction, including thymine-DNA glycosylase (TDG) and tumor necrosis factor (TNF)-related apoptosis-inducing ligand (TRAIL). In addition, *KRAS* mutations are known to induce autophagy in order to promote cell survival and tumor progression through MAPK and PI3K regulation. Thus, these mutations confer resistance to anti-cancer drug treatment and, consequently, result in poor prognosis. Several therapies have been developed in order to overcome KRAS-induced cell death resistance and the downstream signaling pathways blockade, especially by combining MAPK and PI3K inhibitors, which demonstrated promising results. Understanding the involvement of *KRAS* mutations in apoptosis and autophagy regulation, might bring new avenues to the discovery of therapeutic approaches for CRCs harboring *KRAS* mutations.

## 1. Introduction

RAS proteins are a family of small monomeric guanosine triphosphatases (GTPases) that function as transducers of extracellular stimuli to intracellular signaling. RAS proteins regulate important cellular functions, including apoptosis, autophagy, cell proliferation, differentiation, gene expression, migration, invasion and tumor microenvironment (TME) [1,2,3,4,5].

The *RAS* family includes *Harvey (H-)RAS*, *Kirsten (K-)RAS* and *Neuroblastoma (N-) RAS* genes, which are the most frequent oncogenes and one of the most prevalent drivers of cancer. These three *RAS* genes encode four homologous proteins: HRAS, NRAS, KRAS4A and KRAS4B, whose structures and sequences are highly conserved. RAS isoforms share 85–90% sequence homology in the G-domain and diverge mainly at the C-terminal [6,7,8]. The G-domain contains G motifs, which bind directly to GDP or GTP, such as switch I and II and the P-loop. The C-terminal disparities are caused by post-translational modifications (PTMs) that specifically occur in a string of residues termed the hypervariable region (HVR), which is responsible for appropriate membrane localization, interaction and cellular trafficking of the proteins. Therefore, such C-terminal disparities result in different subcellular localization, that can be linked to variations in the diversity or amplitude of signaling [8,9]. In fact, KRAS has a polybasic region required for plasma membrane localization, whereas HRAS and NRAS are palmitoylated; therefore, they are more likely to localize into lipid rafts [9,10]. KRAS4A and KRAS4B are protein products generated through alternative gene splicing of the fourth exon of the *KRAS* gene, which determines the presence or absence of exon 4A. The alternative fourth exon encodes the HVRs responsible for membrane targeting. Thus, KRAS4A is palmitoylated, whereas KRAS4B is not, because it lacks a site of palmitoylation [11]. Additionally, KRAS4A is expressed at low levels, whereas KRAS4B (hereafter referred to as KRAS) is ubiquitously expressed and accounts for 90–99% of all KRAS mRNA forms [12,13].

Despite their high homology, the functions of RAS proteins also differ significantly and do not display redundant roles. This is particularly surprising since the regions of the proteins that interact with downstream effectors are identical to the three RAS isoforms. Therefore, their specific roles may be explained by various others factors, such as cellular context, differential interaction with effectors, compartmentalized signaling and PTMs [14].

Regarding oncogenic *RAS* mutations, they are found in approximately 30% of all human cancers and contribute to important aspects of the malignant phenotype, such as invasion, programmed cell death, deregulation of tumor-cell growth and the induction of new blood-vessels’ formation [3,15,16]. From all human cancers, *RAS* mutations are more frequently found in about 60–90% of pancreatic cancer cases, followed by approximately 30–50% of colorectal cancer (CRC) cases, and between a range of 20% and 30% of lung cancer cases [12,17]. 

### 1.1. The Oncogene KRAS

As a member of the human RAS family, the *KRAS* oncogene encodes a 21 kDa small GTPase, which functions as an on/off switch protein that alternates between an active GTP-bound and an inactive GDP-bound state, by cleaving the terminal phosphate of the nucleotide. The on/off state of KRAS is regulated by GTPase activating proteins (GAPs) and guanine nucleotide exchange factors (GEFs) (Figure 1). GAPs have a GTPase activity responsible for the inactivation of KRAS, through the hydrolysis of GTP. In turn, GEFs facilitate the activation of KRAS, forcing the release of bound GDP and allowing its replacement by GTP. In mammalian cells, KRAS is normally found in its inactive state—GDP-bound [4,18,19]. 

KRAS proteins are activated following the activation of different receptor tyrosine kinases (RTK), such as the epidermal growth factor receptor (EGFR) [12,16]. Upon ligand binding in the extracellular portion, RTKs dimerize, resulting in conformational changes that lead to the autophosphorylation of their intracellular carboxyl-terminal domain. Such phosphorylation stimulates the binding of proteins containing SH2 domains, also known as docking adaptor proteins, to the phosphorylated tyrosine residues, turning RTKs able to recruit GEFs. Grb2 is an example of an adaptor molecule that binds directly to RTKs, or through other adaptor proteins present on growth factor receptors, such as IRS. Grb2 associates to Sos1, which is a GEF, activating the KRAS-RAF-MEK-ERK-MAPK pathway [20,21]. GEFs interact and activate KRAS, promoting the dissociation of GDP and the binding of GTP [22,23,24]. KRAS, in a GTP-bound active state, transduces intracellular signals through other GTPases and kinases, thus linking the presence of extracellular growth factors to intracellular signaling cascades. There are several intracellular signaling cascades activated by KRAS (Figure 1) [25].

Once they are active, KRAS proteins transduce signals across the plasma membrane [12,16], stimulating several effectors by the recruitment and activation of proteins involved in the propagation of signaling from growth factors and other receptors [18,26].

The protein serine/threonine kinase rapidly-accelerated fibrosarcoma (RAF) was the first RAS effector to be characterized and is still the best known. Upon binding to KRAS-GTP, RAF proteins are relocated to the plasma membrane and activated, leading to the initiation of the mitogen-associated protein kinase (MAPK) cascade. This kinase phosphorylates and activates MAPK/ERK kinase (MEK), which subsequently phosphorylates and activates extracellular signal-regulated kinase (ERK). Downstream of this, ERK can regulate numerous transcription factors and cellular functions, such as cell cycle progression, proliferation, autophagy and apoptosis (Figure 1) [16,27]. 

In addition to the MAPK pathway, KRAS can interact with phosphatidylinositol 3-kinase (PI3K), whose activation leads to the phosphorylation of phosphatidylinositol-4,5-diphosphate (PIP2), resulting in the phosphatidylinositol-3,4,5-triphosphate (PIP3) production. This second messenger is able to recruit and activate a large number of proteins containing a pleckstrin homology domain, including phosphatidylinositol dependent kinase 1 (PDK1) and AKT, whose main downstream effector is a mammalian target of rapamycin (mTOR). Such proteins transmit signals regulating cell cycle progression, proliferation, apoptosis, autophagy, migration, invasion and glucose metabolism (Figure 1) [12,16,26,28].

Another important KRAS effector is RAL guanine nucleotide dissociation stimulator (RALGDS), which activates RAS-like (RAL) GTPases and has pro-survival functions and promotes the cell cycle progression. Additionally, KRAS interacts with T-lymphoma invasion and metastasis protein 1 (Tiam1), a Rho family GTPase, which is implicated in the development of RAS-driven tumors, growth transformation, promotion of cell survival, activation of the c-Jun amino-terminal kinase (JNK) mitogen-activated protein kinase and the activation of the NF-kB transcription factor [29]. RAS also binds to PLCε 44, a phospholipase C isoform responsible for the RAS-mediated production of the membrane lipid diacylglycerol (DAG), which results in calcium release and activation of the PKC signaling cascade involved in survival, proliferation, and calcium mobilization (Figure 1) [12,16,29,30].

The combined action of these signaling pathways regulated by this oncogene can lead to several features of malignant transformation, if cells express *KRAS*-activated mutants [16].

### 1.2. KRAS Mutations and Cancer

Of the three RAS isoforms, *KRAS* has the highest mutation rate (86%) and leads to a poor prognosis. Oncogenic *KRAS* mutations are a hallmark of cancer, being a very frequent event in many cancers, including pancreatic cancers (90%), CRCs (30–50%) and lung cancers, especially non-small-cell lung cancer (NSCLC) (15–20%). Such mutations are also present in endometrial cancer, biliary tract malignancies, cervical cancer, liver cancer, bladder cancer, breast cancer and myeloid leukemia [31,32,33].

*KRAS* mutations, featured by single base missense mutations, lead to alterations in the homeostatic balance of GTP and GDP binding, resulting in its constitutively GTP-bound active state, through the reduction in GTP hydrolysis or the increase in the rate of GTP loading. Thus, mutated *KRAS* is able to constitutively activate oncogenic pathways and cellular signal transduction [7,34,35,36]. Point substitutions in codons 12 and 13 are the most common oncogenic *KRAS* mutations, representing 90% of them. In addition, mutations occur less frequently in codons 61, 63, 117, 119 and 146 [32,37]. In detail, the hotspot *KRAS*-mutated codons 12 and 13 correspond to a glycine, and are positioned in the P-loop of KRAS protein, which is essential in maintaining its active form. In these codons, the replacement of glycine by other amino acids, except proline, prevents the arginine finger of GAPs from promoting hydrolysis of GTP [14,21,34,38,39]. Thus, the *KRAS* hotspot mutations result in insensitivity to GAPs increasing time in the GTP bound state [21,22,40,41,42,43,44,45]. In other words, these *KRAS* mutations prevent GAPs from promoting GTP hydrolysis, resulting in the constitutive activation of the KRAS protein and the downstream pathways [24,41,45,46]. Different amino acid substitutions activate different KRAS downstream signaling pathways and display different clonogenic growth potential and responses to targeted therapies. This happens because different mutations influence the way the interaction between KRAS and its effectors occurs [21,45]. Codon 12 mutations increase aggressiveness by the differential regulation of KRAS downstream pathways that leads to the inhibition of apoptosis, the enhanced loss of contact inhibition, and the increased predisposition to anchorage-independent growth. Codon 13 mutations lead to reduced transforming capacity compared to codon 12 mutations [43]. Alternatively, codon 61 substitutions activate KRAS through a similar mechanism, indicating the essential nature of codon 61 in KRAS deactivation [21]. This constitutively active KRAS protein contributes to cell proliferation, suppression of apoptosis, altered cell metabolism and changes in the tumor microenvironment, which leads to tumorigenesis, tumor maintenance, invasion and metastasis [2,24,46,47].

Generally, tumors harboring *KRAS* mutations present higher resistance to chemotherapy and EGFR-inhibitors-targeted therapy, including cetuximab and panitumumab, leading to a worse overall survival, especially in CRC [37,48]. 

In addition, the direct inhibition of KRAS has not presented successful results and efforts have been made to focus on targeting their downstream signaling proteins [49]. This is why it is so important to find the Achilles heel of mutated *KRAS*. KRAS is implicated in the regulation of crucial cellular processes that can prevent tumorigenesis, including apoptosis and autophagy [1,12]. However, there are still several unanswered questions regarding the role of *KRAS* mutations in autophagy and apoptosis regulation, and their regulation loop. Here, we aim to review the role of KRAS in processes, such as autophagy and apoptosis, and the influence of their mutations in apoptosis resistance. Furthermore, we aim to highlight the role of autophagy in cancer and how *KRAS* mutants modulate autophagy. We also intend to explore which therapies have been developed in order to target KRAS-induced cell death resistance.

## 2. KRAS Role on Apoptotic Cell Death

In most human cancer types, apoptosis evasion is an acquired trait, being one of the hallmarks of cancer [50,51]. Apoptosis is a programmed cell death process characterized by caspase activation and morphological features, such as nuclear fragmentation and chromatin condensation [52,53]. Its regulation includes the counterbalance between the pro- (eg., BAX, Bim, Puma, Bad) and anti-apoptotic (eg., Bcl-2, Bcl-xL, Mcl-1) members of the Bcl-2 family. These proteins control death signaling through the regulation of the permeabilization of the mitochondrial outer membrane. This is responsible for the release of apoptotic factors, including cytochrome *c*, which subsequently leads to caspases activation [54,55,56]. In human cancers, apoptosis resistance may be due to the downregulation of pro-apoptotic proteins and/or due to the overexpression of anti-apoptotic Bcl-2 family members. The loss of the TP53 tumor suppressor also constitutes a mechanism for apoptosis resistance and it is the most common one [50,51,57]. Therefore, tumor cells are able to create mechanisms for escaping cell death and, consequently, increasing therapy resistance, meaning that apoptosis induction is essential for tumor regression [58].

As a member of the human RAS family, the *KRAS* oncogene can modulate apoptosis, through the regulation of the downstream effector pathways. The regulation of apoptosis via RAS proteins results in cell survival, mainly through the activation of the PI3K pathway. RAS can bind to and activate PI3K, resulting in AKT activation. AKT is responsible for the phosphorylation of many substrates involved in the regulation of apoptosis. This protein leads to the inhibition of Bad, a pro-apoptotic protein from the Bcl-2 family, preventing the inhibition of the anti-apoptotic proteins Bcl-2 and Bcl-xL [12,59,60]. The activity of caspase 9 and the forkhead-box transcription factors (Fox) are equally inhibited by AKT, resulting in survival. Furthermore, PI3K can activate Rac, which consequently activates another essential transcription factor, the Nuclear Factor-kB (NF-kB). This factor is involved in the production of survival signals, as it promotes the transcription of several anti-apoptotic genes, including inhibitors of apoptotic proteins (IAPs). In addition, Rac can be activated and lead to the activation of NF-kB through Tiam 1, and AKT can phosphorylate and activate the IkB kinase (IKK), thus stimulating NF-kB [59,61,62]. Furthermore, the induction of the MAPK pathway contributes to the control of apoptosis, mostly converging to the same targets as the PI3K pathway [12,59]. Through MEK activity, RAS leads to a downregulation of Par-4, a pro-apoptotic protein [63]. This pathway can also modulate the expression levels of manifold proteins that belong to the Bcl-2 family, leading to the downregulation of the pro-apoptotic member Bim, and to the upregulation of anti-apoptotic proteins, including Bcl-2, Mcl-1 and Bcl-xL [58,64,65,66]. To help the escaping apoptosis, RAS interacts with Bcl-2, one of the best described RAS effectors, resulting in the upregulation of this anti-apoptotic protein (Figure 2) [67].

In contrast, RAS proteins are demonstrated to engage with proapoptotic signaling pathways. The RASSF family of tumor suppressors are the best known proapoptotic RAS effectors [59,68]. RASSF1 and Nore 1 positively regulate RAS-mediated apoptosis. These proteins form homo- and heterodimers and regulate the activity of the serine-threonine kinase Macrophage Stimulating 1 (Mst1). This protein functions as a cleavage target of caspase-3 and stimulates caspase-3 activation. Thereby, RAS binds to RASSF1/Nore/Mst1 complex to induce apoptosis (Figure 2) [59,69,70]. 

Regarding KRAS specifically, the literature is scarce. Rebollo, Pérez-Sala and Martínez-A (1999) have demonstrated that this oncogene interacts with the anti-apoptotic protein Bcl-2. In addition, after being phosphorylated by PKC, KRAS can be associated with Bcl-xL in the outer mitochondrial membrane, where it may stimulate apoptosis [54]. However, the role of mutated *KRAS* in the regulation of the Bcl-2 family proteins is poorly understood.

### Impact of KRAS Mutations in Resistance to Apoptosis

Oncogenic *KRAS* mutations confer resistance to apoptotic stimuli, resulting in anticancer drug treatment resistance and poor prognosis. Therefore, activated *KRAS* is essential to tumor maintenance, as its removal results in apoptosis [59,71]. Previous studies have demonstrated that *KRAS* knockdown results in apoptotic cell death in several *KRAS*-mutant tumor-derived cell lines, showing that some cancer cells require KRAS to maintain viability [72,73]. 

In addition, mutated *KRAS* induces an upregulation of anti-apoptotic Bcl-xL expression, that was modulated by the downstream ERK, in CRC cells. This constitutes a mechanism of apoptosis resistance in *KRAS* mutant CRC [57]. In contrast, *KRAS* knockdown decreases Bcl-xL expression [71]. 

Moreover, ERK activation can induce the expression of the anti-apoptotic proteins, including Bcl-2, Bcl-xL and Mcl-1, promoting survival [66]. In fact, in pancreatic tumor cells, the ERK pathway which is continuously activated by *KRAS* mutations, confers resistance against apoptosis and regulates the progression of these cells in the cell cycle [66]. Thus, oncogenic *KRAS* leads to an increased cell proliferation, resulting in carcinogenesis [48].

Besides Bcl-2 family proteins, oncogenic *KRAS* is associated with the resistance to apoptosis induced by tumor necrosis factor (TNF)-related apoptosis-inducing ligand (TRAIL). This member of the TNF family is involved in apoptosis induction through the binding to its transmembrane death receptors [74]. As these receptors are more abundant on the surface of cancer cells and absent in most normal cells, previous reports show that TRAIL can induce apoptotic cell death in lung and pancreatic cancer cells [75,76]. However, the *KRAS^G12D^* mutation confers resistance to TRAIL-induced apoptosis in pancreatic and lung cancer cell lines. In addition, an inability to induce downstream apoptosis pathways was noted [74].

Furthermore, the mutant *KRAS* is involved in regulation of survivin stability. This protein is one of the members of the IAP family, whose function is associated with cell cycle progression and apoptosis inhibition [77,78,79]. Survivin levels in normal cells are very low or absent, whereas in cancer cells, very high levels are observed. Interestingly, *KRAS* depletion leads to a decrease in survivin levels in cancer cells harboring *KRAS* mutants, but not wild-type *KRAS*. This process occurs through the induction of ubiquitination and proteasomal degradation of this IAP protein, by KRAS. In addition, survivin depletion compromises the ability of oncogenic *KRAS* to promote invasion, anchorage-dependent growth and survival. Therefore, a *KRAS*-driven malignant transformation can be dependent on high levels of survivin [78]. 

Suppression of apoptosis by *KRAS* mutations can also be associated with the downregulation of thymine-DNA glycosylase (TDG). This protein has a crucial role in DNA demethylation, and can restore the sensitivity to apoptosis, through the recruitment of the histone lysine demethylase JMJD3 to Fas promoter. Downregulation of TDG is verified in pancreatic cancer cells with *KRAS* mutations, in contrast to those expressing the wild-type isoform of *KRAS*. Furthermore, TDG expression levels are restored after *KRAS* knockdown [80,81]. 

In addition, oncogenic *KRAS* can affect the expression of the microRNA 200 (mir-200) family, which comprises five members: mir-200b, mir-200 a, mir-429, mir-200c and mir-141 [82,83]. This microRNA family is involved in apoptosis regulation and their expression levels are deregulated in tumor cells. Downstream effectors of KRAS, JUN and SP1, through the RAF/MEK/ERK pathway, are involved in mir-200 inhibition, thus repressing its function as a tumorigenesis suppressor. Moreover, mir-200 can directly repress the anti-apoptotic protein Bcl-2 by abrogating the resistance to apoptosis induced by KRAS [82,83]. These small non-coding RNAs are associated with the induction of pro-tumor autophagy pathways [1,84]. Interestingly, mir-200b is also involved in autophagy regulation [85]. Pan and co-workers observed a negative correlation between mir-200b and autophagy-associated gene 12 (ATG12) and that mir-200b downregulated ATG12, inhibiting autophagy [85]. In addition, mir-200c-3p overexpression induces autophagy through the activation of light chain-3 (LC3)-II and the formation of autophagosome in PC-3 prostate cancer cells [86]. Thus, the mir-200 family is a clear example of the complex signaling network that connects the *KRAS* oncogene with a diversity of cellular processes and proteins, such as apoptosis and autophagy.

Overall, we conclude that mutated *KRAS*-induced resistance to apoptosis is a result of a complex signaling network that connects this oncogene with a diversity of cellular processes and proteins.

## 3. Role of Autophagy in Cancer

Autophagy is a catabolic process that involves the engulfment of intracellular components from cytosol to lysosome or vacuole, in animal cells or in yeast and plant cells, respectively, for their degradation. Autophagy is crucial for homeostasis and is not only important for the disposal of damaged proteins and organelles, but also constitutes an adaptive response to several stresses, providing energy and nutrients [85,87,88]. In mammalian cells, this “self-eating” process begins in the cytosol with the formation of a double-membrane autophagosome. The process of autophagosome formation is regulated by several proteins, including Beclin-1 and autophagy related genes (ATGs). After autophagosome maturation, which includes the fusion with a lysosome, forming an autolysosome, and/or with an endosome, forming an amphisome, the degradation of vesicle contents occurs through lysosomal hydrolases. The resulting molecules are recycled through permeases [87,89]. 

In cancer cells, autophagy plays an ambiguous role—evasion of cell death, favoring stress adaptation, or cell death, destructing the cell—depending on context and tumor cell specificity. Thereby, autophagy can be involved in both tumor prevention and in tumor initiation or promotion [90,91]. Several factors can stimulate autophagy to promote cell survival, including glucose and oxygen deprivation, amino-acid starvation, cytotoxic cellular damage and growth-factor withdrawal [92]. Under nutrient-deprived conditions and hypoxia, autophagy can provide oxygen and nutrients through the recycled cellular components, promoting survival. Therefore, in solid tumors, cells that are centrally located lack access to oxygen and nutrients and present increased levels of autophagy [91]. This means that energy and glucose depletion are activators of autophagy to promote survival [1]. Consistent with this, in tumor cells defected in apoptosis, this mechanism increased survival. Furthermore, this mechanism plays an important role in invasion, motility, proliferation and metastasis [52]. In contrast, in the early stages of tumor development, autophagy can prevent tumor formation and act as a tumor suppressor, by limiting genome instability, inflammation and tissue damage. In this case, its inhibition and defects on this process are associated with the continuous growth of pre-malignant cells and increased tumorigenesis [90,93]. Because of this, it appears that autophagy inhibits tumor progression in the early stage and helps the tumor growth in late stages where the metabolic demands are higher. To confirm this, previous studies demonstrated that, under starvation conditions, autophagy and mutated *KRAS* contribute to CRC cell survival. Furthermore, inhibiting glycolysis and oxidative phosphorylation may induce autophagy in *KRAS*-mutant human CRC cell lines, as a pro-survival process [1,94].

The role of autophagy can also differ between the different cancer cell types. In breast cancer, autophagy is activated to maintain amino acid levels under nutrient starvation conditions. In turn, in CRC cells, this process is crucial for growth and proliferation, even under conditions of available nutrients [95]. Furthermore, the involvement of key regulators of autophagy can differently influence cancer cell fate, also depending on cancer type. Beclin-1 expression can be related to tumor suppression, having an inhibitory role on cell proliferation in several cancer types, including hepatocellular carcinoma, tongue squamous cell carcinoma, lung cancer, breast cancer, cervix cancer, CRC, pancreatic cancer, glioblastoma and squamous cell carcinoma [96]. In contrast, Beclin-1 can promote tumorigenesis in a context dependent manner, as its complete knockout in triple negative breast cancer cells leads to cell cycle arrest, resulting in impaired tumor growth [96,97]. ATGs are also relevant in human cancers as mutations in these genes are found in gastric cancer, CRC and hepatocellular carcinoma. Such mutations lead to autophagy deregulation and cancer development [84]. 

Overall, autophagy is an issue to be concerned as a potential mechanism of resistance to anticancer agents, as it aids in the response of tumor cells to cellular stress and/or increased metabolic demands [1]. 

### Modulation of Autophagy by KRAS Mutations

In normal cells, RAS proteins are involved in autophagy regulation mainly through the PI3K/AKT pathway, which controls mTOR expression and suppresses the autophagic process. In addition, RAS proteins can regulate the expression levels of the key regulator of autophagy, Beclin-1, decreasing it and inhibiting autophagy. Oppositely, RAS can lead to autophagy induction through the MAPK pathway, which is activated by amino acid starvation in CRC [12,86,98], and through the increase in the expression of essential constituents of the autophagy machinery, including ATG5 [99].

Regarding RAS-driven cancer cells, they generally present high levels of autophagy, to preserve the mitochondrial function [100]. RAS-induced autophagy can support the adaptation of cancer cells to the challenging microenvironment through the supply of energy and nutrients, promoting cell survival [1,101]. Additionally, in the presence of mutated RAS, autophagy may induce an adhesion-independent transformation and stimulate glycolysis (Figure 3). Furthermore, autophagy is essential in RAS-driven transformations, and tumor cells with mutated RAS are particularly dependent of this mechanism [17,102]. More specifically, the mutation *KRAS^G12V^* was described as contributing to the upregulation of autophagy and, consequently, cell proliferation [90,100,102]. 

On the other hand, some reports have shown that knockdown of autophagy in cells with oncogenic RAS may promote tumor growth, which supports the idea that RAS-driven autophagy also leads to cell death [103]. Moreover, the activation of MAPK pathways leads to the upregulation of a member of the Bcl-2 family, Bcl-2/adenovirus E1B 19 kDa protein-interacting protein 3 (BNIP3), which induces autophagy and promotes CRC cell death [104]. Oncogenic RAS can also induce autophagic cell death, through the upregulation of Beclin-1 and Noxa, a BH3-only protein, member of the Bcl-2 family. Noxa or Beclin-1 silencing increases cell survival and decreases RAS-induced autophagy. Through this mechanism, the oncogenic potential of mutated RAS is reduced [105]. In CRC, autophagy can promote cell survival under starvation conditions and cell lines harboring *RAS* mutations have high levels of autophagy [1,85,106]. Previous studies reported that, under nitrogen starvation conditions, the expression of activating *KRAS* mutations increases Atg8p levels in yeast, which is an autophagic marker. Consistently, KRAS-induced autophagy supports the survival of CRC-derived cells exposed to stressful conditions, such as the limitation of nutrients [1]. Moreover, KRAS-induced autophagy is mediated through an upregulation of the MAPK pathway and downregulation of the PI3K/AKT pathway, known to activate the autophagy inhibitor mTOR (Figure 3) [1,103]. 

Regarding pancreatic ductal adenocarcinoma, autophagic flux present in these cells can be involved in tumor maintenance as *KRAS*-mutant cells demonstrate high basal levels of autophagy [107,108]. Remarkably, mutant *KRAS* suppression leads to an increased autophagic flux rather than decreased basal levels, and the chronic ablation of *KRAS^G12D^* results, in mouse pancreatic ductal carcinoma cells, in more dependence on autophagy. In addition, ERK inhibition has the same phenotypic effect of *KRAS* suppression, resulting in autophagy stimulation. This enhanced effect of autophagy increases autophagosome flux, and the transcription of autophagy-related genes leads to activation of AMPK and Beclin-1 and downregulation of the mTOR pathway [107]. 

In a *KRAS^G12D^*-driven NSCLC mouse model, ablation of *ATG7* leads to a suppression of proliferation, thus decreasing tumor growth. Furthermore, *ATG7* deficiency results in an accumulation of defective mitochondria and affects tumor fate, as adenomas and carcinomas become more benign oncocytomas. These results corroborate the concept that autophagy promotes tumor growth in *KRAS*-driven cancers [109,110]. 

Overall, *KRAS* mutations induce autophagy to promote survival and tumor progression. Thus, inhibition of KRAS or autophagy could be a promising therapeutic strategy for tumor cells harboring *KRAS* mutations. In fact, co-targeting autophagy and the MAPK pathway may be a potential therapeutic approach for *KRAS*-mutant cancers [109,111].

## 4. The Autophagy/Apoptosis Regulation Loop

Several studies reported the existence of a crosstalk between apoptosis and autophagy, because apoptosis regulators are also involved in the control of autophagy and both processes are often activated together to respond to stress stimuli (Figure 4) [112,113]. 

Members of the Bcl-2 family proteins may have influence in autophagy regulation. Beclin-1 has a BH3 domain that binds to the BH1-BH2 domains of Bcl-2, Bcl-xL or Mcl-1 [85]. This interaction results in autophagy inhibition and is not obligate, as it only occurs in certain conditions, when there are abundant nutrients [85,114,115]. Under nutrient starvation conditions, Bcl-2 is phosphorylated by JNK1, preventing its binding to Beclin-1, allowing the initiation of autophagosomes formation [116]. Furthermore, Bcl-2 and Bcl-xL stimulate a cytoprotective autophagy, a role that is independent from BAX, and, in apoptosis-deficient cells, an overexpression of Bcl-2 may potentiate autophagy. Remarkably, these anti-apoptotic proteins may have anti or pro-autophagic functions. In addition, Bcl-xL and ATG7 may belong to the same signaling pathway, because the downregulation of this anti-apoptotic protein is balanced by ATG7 [85]. BAX seems to have a negative influence on autophagy, which is dependent on caspase-3 activity, as it leads to Beclin-1 cleavage. In turn, this cleaved autophagic protein stimulates apoptosis, increasing the release of cytochrome *c*. Furthermore, BAX inhibits the autophagosome synthesis, a process that can be reverted by Bcl-xL (Figure 4) [117,118].

In addition, some perturbations in the apoptotic machinery may induce autophagic cell death. The inhibition of caspase or cysteine proteins results in the blockade of apoptosis and consequently in autophagic cell death. Furthermore, the knockdown of the pro-apoptotic proteins BAX and BAK has the same consequence [112]. Remarkably, in the absence of these pro-apoptotic proteins, the expression levels of Bcl-2, Bcl-xL and Mcl-1 have no effect on autophagy. However, in the presence of BAX and BAK, the inhibition of such anti-apoptotic Bcl-2 family members leads to autophagy stimulation, which is associated with increased apoptotic cell death. Therefore, the pro-survival Bcl-2 family members influence the autophagic process indirectly, through the activation of BAX and BAK (Figure 4) [119]. 

Regarding *ATG* genes, these may regulate the interaction between autophagy and apoptosis. Calpain-cleaved ATG5 and Bcl-2 interaction facilitates apoptosis. Moreover, ATG7 suppression leads to autophagy inhibition, suppression of caspase activation and decreased cell death. Therefore, in apoptosis-deficient cells, the downregulation of ATG7 and ATG5 may suppress cell death [113,120]. The conjugation ATG12-ATG3 can also regulate apoptosis, as its disruption may result in increased Bcl-2 expression, suppression of cell death and mitochondrial mass expansion [120]. Moreover, overexpression of ATG1 can stimulate autophagy in *Drosophila melanogaster*, and autophagy activation inhibits cell growth and leads to apoptotic cell death (Figure 4) [121]. 

Furthermore, p53, a known activator of apoptosis, can regulate autophagy and its cellular localization influences its effect on autophagy [112,120]. TRAIL, a classic ligand involved in apoptosis induction, may also induce autophagy, in a model of lumen formation in mammary acini. In contrast, TNF-α can both induce and repress autophagy. Induction of autophagy occurs in Ewing sarcoma cells lacking NF-kB expression, leading to ROS-dependent Beclin-1 overexpression and induction of apoptosis. Furthermore, in the absence of NF-kB activation, knockdown of *ATG* genes decreases apoptosis induced by TNF-α. On its turn, NF-kB activation results in autophagy suppression in TNF-α-treated Ewing sarcoma cells, which is correlated with NF-kB-mTOR activation (Figure 4) [122]. 

Autophagy and apoptosis can also interact at the level of the signaling pathways. AKT and ERK activate pathways that influence apoptosis and autophagy, including Bcl-2 family members and mTOR [113]. Upstream activators of mTOR mediate its activity in a caspase-dependent manner [120]. 

In conclusion, there is a crosstalk between autophagy and apoptosis, but they establish different types of interplay. Autophagy can play an anti- or a pro-apoptotic role, depending on the cellular context [113,122]. However, the precise mechanisms that regulate this crosstalk remains to be elucidated. 

## 5. Therapies Targeting KRAS-Induced Cell Death Resistance

Despite the well-recognized importance of KRAS in cancer and the extensive efforts to develop therapies against its mutant, KRAS has been considered to be undruggable [32,123]. Firstly, *KRAS* mutations were identified as a predictive biomarker for anti-EGFR therapy inefficacy [123,124]. Although EGFR antibodies bind to EGFR with high specificity, mutations in downstream cascade, including *KRAS*, lead to the constitutive activation of the intracellular signaling pathway, independently of the stimulation via EGFR. Thus, in such a situation, it is redundant to inhibit this receptor [124,125]. Nevertheless, EGFR stimulates the PI3K pathway, leading to autophagy inhibition. Therefore, anti-EGFR therapy can induce autophagy. Panitumumab was reported to increase Beclin-1 protein levels, which probably constitutes a protective mechanism. Patients treated with cetuximab present increased levels of autophagy-related proteins. Moreover, autophagy activation through anti-EGFR agents can be related to a cytoprotective role in solid tumors and several cancer cell lines [126,127,128].

Nowadays, several therapies have been developed in order to directly or indirectly target KRAS, through downstream signaling pathways blockage, KRAS synthetic lethal interactors, KRAS plasma membrane association inhibitors, post-translational modifications, KRAS-regulated metabolic processes, KRAS-mediated inflammation and immunotherapy (Figure 5) [32,129,130,131,132,133,134,135,136]. 

Inhibitors that directly target *KRAS* mutations overall were considered “mission impossible”, due to KRAS’ picomolar affinity to GTP and high intracellular GTP concentration [32,137]. In fact, the difficulties to target the GTP binding pocket, as well as the lack of well-defined hydrophobic pockets on the RAS protein surface, fail to obtain effective small molecules that directly bind to the KRAS GTP-binding site [32,137]. However, the discovery that *KRAS^G12C^* mutation has cysteines in its active sites, which are not present at wild-type or other mutated *KRAS*, enables that this mutation may be specifically inhibited by covalently targeting its active site of cysteines [138]. Thus, a novel direct inhibition of mutated *KRAS* emerges as one of the most promising strategies that specifically bind in a covalent and irreversible way to the mutated *KRAS^G12C^*, and favor KRAS-GDP state over GTP, due to the decrease of affinity in RAS for GTP compared with GDP [32,123,125,135,136,137]. These phenomena inhibit not only RAF binding but also downstream signaling pathway activation, decreasing viability and increasing apoptosis of cancer cells harboring *KRAS^G12C^* mutations [123,125,135,136]. Several inhibitors have already been developed, such as ARS-853, ARS-1620, AMG-510 (Sotorasib) and MRTX849 (Adagrasib), with AMG-510 and MRTX849 being the first ones to enter the clinic for non-small cell lung cancer (NSCLC) (Figure 5) [32,123,125,137]. Unfortunately, although *KRAS^G12C^* targeting has a high relevance on lung cancer due to its high incidence, it has a marginal impact in CRCs in which the mutation has a low incidence compared with *KRAS^G12D^* (11%), *KRAS^G13D^* (7%) and *KRAS^G12V^* (7%) (versus *KRASS^G12C^* (4%)) [125]. Thus, it is imperative to target other *KRAS* mutations [138].

The downstream signaling pathways blockage, in particular the MAPK and PI3K pathways, have been also explored. Although MEK inhibitors are demonstrated to be effective in blocking MEK activation via RAF-mediated phosphorylation, unexpectedly, this target therapy results in the activation of the other arm of RAS signaling, particularly PI3K [139]. In its way, RAS-transformed human cells sustain their tumorigenic activity through the PI3K pathway [139]. In pancreatic ductal adenocarcinoma, several studies show that combining an MEK inhibitor, such as AZD6244, with PI3K inhibitor BKM120 or GDC0941, results in increased apoptosis, especially in pancreatic cell lines resistant to single agents [139]. Additionally, some ERK inhibitors (BVD-523 and LY3214996) combined with nab-paclitaxel and gemcitabine (nucleoside analog) are currently under evaluation in phase I clinical trials, in pancreatic ductal adenocarcinoma [130]. The combination of MEK inhibitor (trametinib) with nab-paclitaxel results in an increased expression of apoptosis-related proteins, including cleaved caspase-3 and cleaved PARP-1, in pancreatic cancer models [140]. In CRC cells, targeting PI3K/mTOR signaling through its inhibitor NVP-BEZ235 also seems to be an effective strategy, as it induces growth inhibition and apoptosis through the upregulation of Bim expression (Figure 5). This inhibitor has also shown effectiveness in other types of cancer, including breast cancer, multiple myeloma, and sarcoma [141,142]. In contrast, in NSCLC cells, the development of resistance to MEKi-PI3Ki therapy was observed. However, combining this therapy with Bcl-2 family proteins inhibitors, such as ABT-263 (navitoclax), may be an useful strategy, as an apoptosis stimulation was verified [58]. Furthermore, combinatory strategies to enhance the apoptotic activity of Rafi-MEKi-ERKi and to overcome the ineffectiveness and toxicity of single agent pharmacological inhibition have been explored. The most potent combination seems to be the treatment using a pan-RAFi together with an ERK-selective inhibitor (RAFi/ERKi). Interestingly, this strategy abrogates the strong induction of compensatory signaling that can drive to the ERK reactivation, causing cell cycle arrest and apoptosis, impaired metabolism and loss of MYC-, E2F- and FRA1-dependent transcriptomes. The same strategy was also investigated in CRAF mutant melanoma, specifically with BRAFi/MEKi therapy, and is now approved in the clinic. It was observed that this combinatory strategy delays resistance and reduces toxicity comparing with single-agent treatment alone. The same was also observed in the RAFi/MEKi strategy [143]. The inhibition of the downstream transcription factor Fos-like antigen 1 (FOSL1) also seems to be a promising target in *KRAS* mutant lung and pancreatic cancers [32]. Furthermore, the inhibitor of mTOR, AZD8055, combined with the dual BCL-XL/BCL-2 inhibitor, ABT-263, demonstrates promising results (Figure 5). This approach induces robust apoptosis in *KRAS* mutant human CRC cell lines, but not in KRAS wild-type CRC models. Additionally, tumor regression is also observed in *KRAS* mutant CRCs in vivo both in human xenografts and in wild-type (APC/p53) genetically-engineered mouse models (GEMMs) of CRC [144]. 

An alternative approach is the KRAS synthetic lethal interactors that are based on the targeting of co-dependent vulnerabilities or synthetic lethal partners that are essential for *KRAS* oncogenesis [135]. Thus, several efforts have been made in order to identify these secondary targets whose loss of function would be uniquely lethal in the presence of mutant *KRAS*, but not in the presence of wild-type KRAS [32,137,139]. SCR homology region 2-containing protein 10 tyrosine phosphatase 2 (SHP2) was identified as one synthetic lethal interaction partner by screening methods using siRNA, shRNA, RNAi and CRISPR library screens [137]. It was observed that its inhibition induces the vulnerability of *KRAS* mutant in NSCLC [137].

Regarding KRAS plasma membrane association inhibitors and KRAS post-translational modifications, both intend to modulate KRAS membrane association and their biological activation. Therefore, farnesyl-transferase inhibitors (FTI) were developed in order to prevent prenylation of the CAAX cysteine, which is required for oncogenic transformation [32,134]. Although a large number of highly effective FTIs have been identified, phase I and II trials demonstrate poor clinical efficacy [32,134,136]. In fact, KRAS4B can also be modified by geranylgeranyl transferase (GGT), supporting KRAS bioactivity when farnesylation is impaired [32,136]. Thus, FTI and GGT combined therapy has been explored. However, besides the efficacy of combined therapy in reducing *KRAS*-driven lung tumorigenesis in mice, high toxicity in normal tissues was also observed (Figure 5) [32,136]. 

Due to recent findings that oncogenic *KRAS* can promote a metabolic reprogramming of tumor cells, several attempts to target KRAS-regulated metabolic process have also been developed [32]. In fact, *KRAS* mutant colon tumors are demonstrated to be associated with an increased expression of glutamine and glycolytic metabolic proteins, while *KRAS* mutant pancreatic tumor showed reprogrammed glutamine metabolism [32]. However, these strategies have been tested in pre-clinical cancer models and still need clinical validation, such as AZD2965, a MCT-1-specific inhibitor that blocks the lactate efflux and, consequently, its toxic accumulation in tumor cells inhibits pancreatic tumor growth [32,139]. It was already reported that inhibition of the MAPK pathway, through inhibitors against *KRAS^G12C^*, ERK and MEK, promotes increased autophagy levels. Due to this observation, several studies are testing the inhibition of autophagy with hydroxychloroquine in combination with inhibitors of the MAPK pathway, and promising results in preclinical models of pancreatic ductal adenocarcinoma and *NRAS*-mutant melanoma were achieved. Indeed, the therapy with hydroxychloroquine plus trametinib (MEK inhibitor) is under evaluation in patients with pancreatic ductal adenocarcinoma and metastatic melanoma, in a phase II clinical trial [138,145]. The efficacy of this autophagic inhibitor has also been investigated, in a phase I/II clinical trial, in combination with binimetinib (MEK inhibitor), ulixertinib (ERK inhibitor) and gemcitabine, in pancreatic cancer and gastrointestinal adenocarcinomas [145,146]. Furthermore, chloroquine was tested in clinical trials in patients with pancreatic cancer, having favorable results, as the blockage of autophagy and the impairment of tumor growth were observed [147,148].

Targeting immune-checkpoint molecules, such as programmed cell death 1 (PD-1), programmed cell death ligand 1 (PD-L1) and cytotoxic T-lymphocyte-associated protein 4 (CTLA-4), has also demonstrated to be one of the most promising cancer treatments, with positive results in *KRAS*-mutated cancers [133,134]. In lung cancer, anti-PD-1/PD-L1 therapies have already been approved, as in advanced-stage NSCLC patients, monoclonal antibodies that target PD-1 and its main ligand PD-L1 have shown survival increments [129,149]. Contrarily, anti-CTLA-4 therapy did not present encouraging results in lung carcinoma [133]. In CRC, anti-PD-1/PD-L1 therapy, such as pembrolizumab, has also been approved in a subgroup of patients, namely mismatch repair deficient (dMMR) ones, which present higher PD-L1 expression when compared with MMR proficient (pMMR) carcinomas [125,133,150,151,152]. In addition, nivolumab is also an FDA approved PD-blocker with good results in dMMR and MSI-H metastatic CRC [152]. In pancreatic cancer, immunotherapy is not included in the clinical guidelines due to the limited clinical success in this type of cancer [133,136]. It was already described that immune-checkpoint inhibitors, as well as anti-EGFR antibodies, trigger autophagy in BRAF^V600E^ CRC cells, being appointed as a potential resistance mechanism to these agents [153]. Thus, co-targeting autophagy seems to be a promising approach to overcome resistance issues [153]. In fact, Koustas and co-workers [153] verified that the co-inhibition of immune-checkpoints, EGFR and autophagy attenuated tumor growth. However, neither immune-checkpoint inhibitors nor anti-EGFR antibodies, alone or in combination, triggered autophagy in MSI-H CRC cell line HCT116 harboring *KRAS*^G13D^ mutation. Moreover, the co-target of autophagy and MEK, alone or in combination with immune-checkpoint inhibitors or with anti-EGFR antibodies, also had no effect on the death of HCT116 cells. In conclusion, anti-EGFR antibodies and immune-checkpoint inhibitors can trigger autophagy in mutated *BRAF*, but not in mutated *KRAS* CRC cells [153].

In conclusion, several strategies have been developed in order to target *KRAS* mutations (Table 1). Combined strategies are shown to be the most successful because of the redundancy of the signaling pathways. However, the cancer type and tumor cells mutations influence the efficacy of the cancer treatment. 

## 6. Conclusions

Oncogenic *KRAS* mutations can regulate important cellular functions involved in malignant transformations through their downstream signaling pathways, including apoptosis and autophagy. The connection between mutated *KRAS* and a diversity of proteins involved in the apoptotic process regulation results in cell death resistance. This interaction promotes an upregulation of the anti-apoptotic proteins and a downregulation of the pro-apoptotic ones. *KRAS* mutations also have a crucial role in autophagy stimulation to induce cell survival and tumor progression. However, the role of mutated human *KRAS* in apoptosis and autophagy regulation is not well understood. Considering the high incidence of *KRAS* mutations and the poor prognosis associated with them, it is urgent to find new approaches for the treatment of cancers harboring *KRAS* mutations, and elucidating *KRAS* mutations involvement in apoptosis/autophagy regulation may contribute to such an aim.

To summarize, in this review we describe the state of the art role of KRAS in apoptosis and autophagy regulation, highlighting the downstream signaling pathways involved, the regulation loop of both processes and how *KRAS* mutations can induce apoptosis resistance and modulate autophagy. In cancer cells, *KRAS* mutations are able to influence the expression levels of a diversity of proteins involved in apoptosis regulation and can also stimulate autophagy in order to favor cell survival and tumor progression. These abilities conferred by *KRAS* mutations constitute an obstacle to anti-cancer therapies, leading to intensive efforts to develop new strategies to directly or indirectly target KRAS. In this review, it is described which new therapies have been developed in order to target KRAS-induced cell death resistance, expecting to bring new avenues to the discovery of novel therapeutic approaches for CRC, based on the involvement of *KRAS* mutations in apoptosis and autophagy regulation.

## Figures and Tables

**Figure 1 cells-11-02183-f001:**
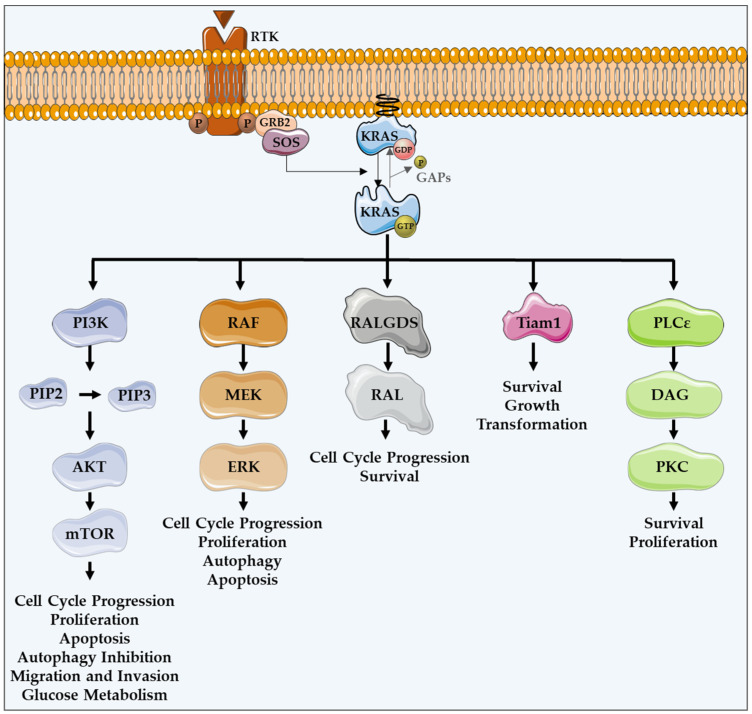
KRAS major effectors and cellular functions. KRAS proteins are activated upon activation of an RTK-like EGFR. KRAS is a GTPase that functions as an on/off switch that alternates between an active GTP-bound and an inactive GDP-bound state, regulated by GAPs and GEFs, such as Sos1. This oncogene regulates several effector pathways, including the MAPK and PI3K/AKT pathways. GTP-bound KRAS leads to the activation of RAF proteins, resulting in the initiation of MAPK signaling. Subsequently, MEK is activated and, in turn, phosphorylates and activates ERK. Downstream of this, ERK can regulate numerous transcription factors, promoting cell cycle progression and influencing proliferation and apoptosis. Besides MAPK pathway, KRAS can interact with PI3K, whose activation leads to the phosphorylation of PIP2, resulting in PIP3. This second messenger is able to activate a large number of proteins containing a pleckstrin homology domain, including PDK1 and AKT, whose main downstream effector is mTOR. These proteins regulate cell cycle progression, cell survival, glucose metabolism, cell growth and proliferation. In addition to MAPK and PI3K/AKT pathways, KRAS can activate RALGDS, whose downstream effector is RAL GTPases, promoting cell cycle progression and survival. Furthermore, KRAS interacts with Tiam1, a Rho family GTPase, which is implicated in the development of RAS-driven tumors, growth transformation and promotion of cell survival. KRAS also binds to PLCε 44, a phospho-lipase C isoform responsible for KRAS mediated production of DAG, resulting in calcium release and activation of the PKC signaling cascade, involved in survival, proliferation, and calcium mobilization.

**Figure 2 cells-11-02183-f002:**
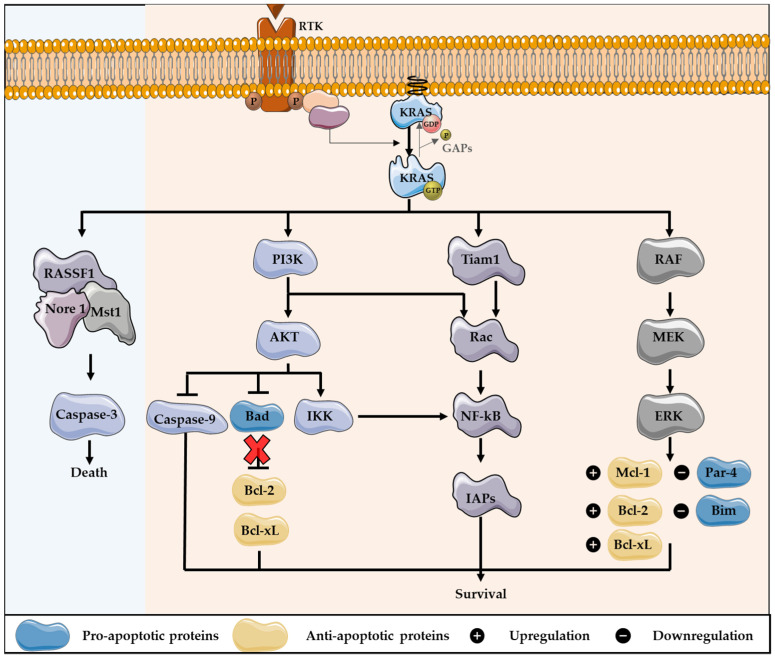
KRAS downstream signaling activation cross-talk with apoptosis pathways regulation. KRAS can modulate apoptosis, through the regulation of the downstream effector pathways, which mainly results in cell survival, especially through the activation of PI3K pathway. After, KRAS activates PI3K and AKT: the last protein leads to the inhibition of Bad, a pro-apoptotic protein of the Bcl-2 family, preventing the inhibition of the anti-apoptotic proteins Bcl-2 and Bcl-xL. Furthermore, PI3K can activate Rac, which consequently activates the transcription factor NF-kB, involved in the production of survival signals, as NF-kB promotes the transcription of several anti-apoptotic genes, including inhibitors of apoptotic proteins (IAPs). In addition, Rac can lead to the activation of NF-kB through Tiam 1, as also AKT can phosphorylate and activate IKK, thus stimulating NF-kB. Additionally, the induction of MAPK pathway contributes to the control of apoptosis, through MEK activity. Thus, KRAS leads to the downregulation of Par-4, a pro-apoptotic protein, and the expression level of several proteins that belong to Bcl-2 family is regulated. This includes the downregulation of Bim, a pro-apoptotic member, and the upregulation of the anti-apoptotic proteins Bcl-2, Bcl-xL and Mcl-1. In contrast, KRAS can engage pro-apoptotic signaling pathways, through the activation of RASSF1/Nore 1/Mst1 complex, stimulating caspase-3 activation.

**Figure 3 cells-11-02183-f003:**
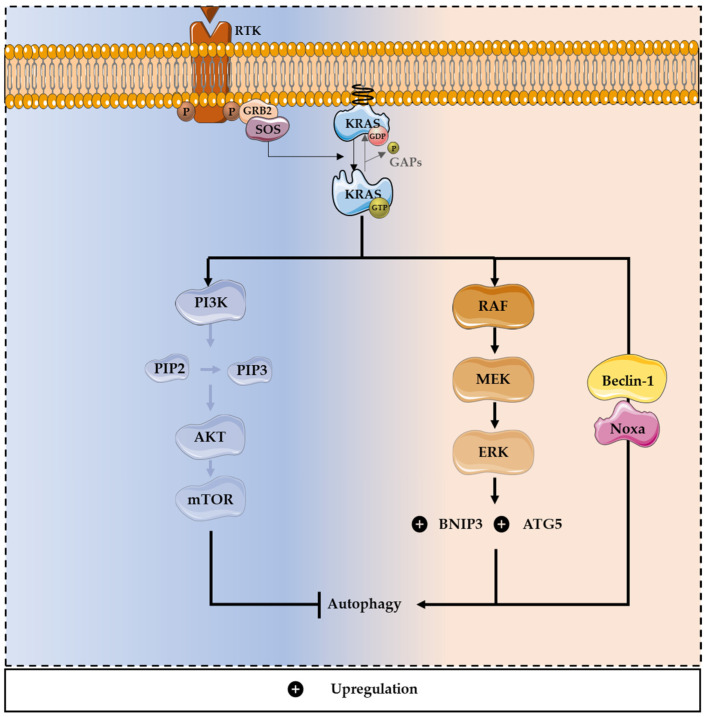
Dual role of mutated KRAS in autophagy modulation. KRAS-induced autophagy is mediated through upregulation of the MAPK pathway and downregulation of the PI3K/AKT pathway, promoting survival and tumor progression. Through the MAPK pathway, oncogenic *KRAS* can promote autophagy, enhancing the expression of essential constituents of the autophagy machinery, including ATG5. The activation of the MAPK pathway also leads to the up-regulation of a member of the Bcl-2 family, BNIP3, inducing autophagy and promoting CRC cell death. Moreover, oncogenic *KRAS* can induce autophagic cell death, through the upregulation of Beclin-1 and Noxa, a BH3-only protein and a member of the Bcl-2 family.

**Figure 4 cells-11-02183-f004:**
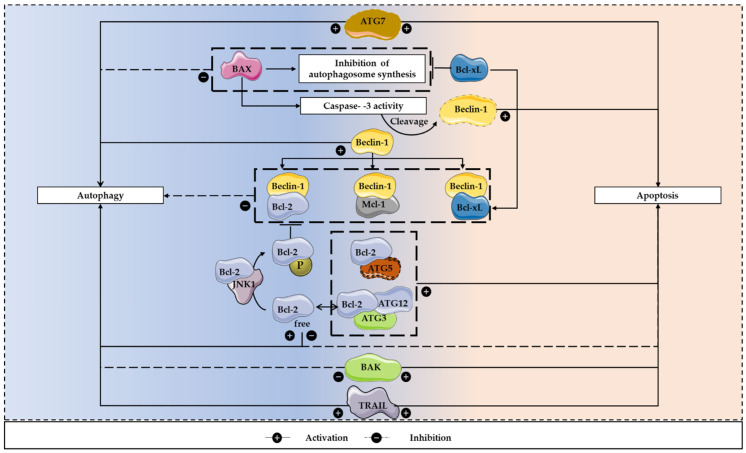
The apoptosis/autophagy regulation loop. There are several apoptosis regulators that are also involved in the control of autophagy and both processes are often activated together to respond to stress stimuli. These interactions can result in both inhibition and induction of autophagy. To inhibit the autophagic process, Beclin-1 can bind to Bcl-2, Bcl-xL and Mcl-1. In contrast, under nutrient starvation conditions, Bcl-2 is prevented in binding to Beclin-1, after being phosphorylated by JNK1, allowing the initiation of autophagosomes formation. Furthermore, BAX can lead to Beclin-1 cleavage, which is dependent on caspase-3 activity. This pro-apoptotic protein can also inhibit the autophagosome synthesis, a process that can be reverted by Bcl-xL. In addition, activators of apoptosis, including TRAIL, can regulate autophagy, having a positive influence on this mechanism. Regarding apoptosis induction, calpain-cleaved ATG5 and Bcl-2 interaction facilitates this programmed cell death process. Moreover, ATG7 expression leads to autophagy induction, caspase activation and increased cell death. The conjugation ATG12-ATG3 can also regulate apoptosis, as its disruption may result in increased Bcl-2 expression and decreased apoptotic cell death.

**Figure 5 cells-11-02183-f005:**
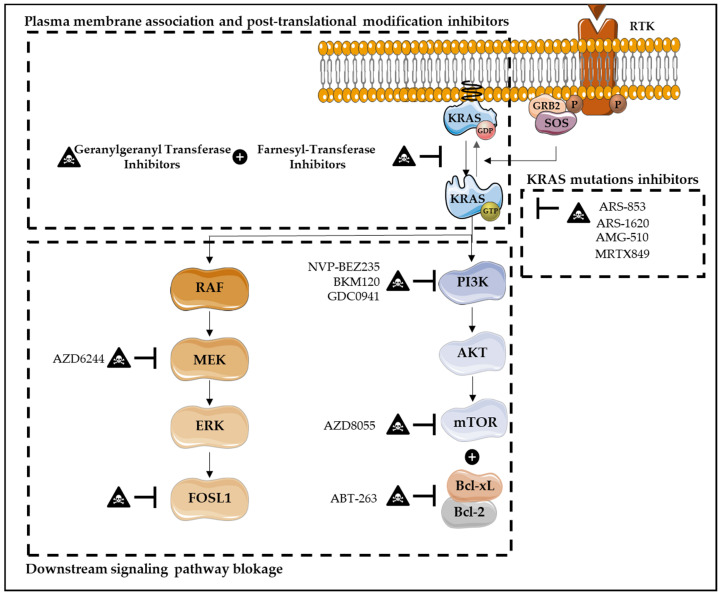
Therapies targeting KRAS-induced cell death resistance. Several therapies have been developed in order to target KRAS, including downstream signaling pathways blockade, direct KRAS inhibitors, KRAS plasma membrane association inhibitors and post-translational modifications. Downstream signaling pathways blockade, in particular MAPK and PI3K pathways, has also been explored. The inhibition of the downstream transcription factor FOSL1 seems to be a promise target in KRAS mutant lung and pancreatic cancers. Additionally, the inhibition of mTOR, AZD8055, combined with the dual BCL-XL/BCL-2 inhibitor, ABT-263, has demonstrated promising results. This approach induces robust apoptosis in *KRAS* mutant human CRC cells, but not in KRAS wild-type CRC models. In CRC cells, targeting PI3K/mTOR signaling, through its inhibitor NVP-BEZ235, also seems to be an effective strategy, as it induces growth inhibition and apoptosis. Similarly, in pancreatic ductal adenocarcinoma, combining MEK inhibitors, such as AZD6244, with the PI3K inhibitors BKM120 or GDC0941 results in increased apoptosis. Regarding direct KRAS inhibitors, these are one of the most promising strategies, in which KRAS-GDP state is favored over KRAS-GTP binding. Several inhibitors have already been developed, such as ARS-853, ARS-1620, AMG-510 and MRTX849, with AMG-510 and MRTX849 being the first ones to enter the clinic. Furthermore, KRAS plasma membrane association inhibitors and KRAS post-translational modifications have been developed, with the intention to modulate KRAS membrane association. Farnesyl-transferase inhibitors and geranylgeranyl transferase combined therapy have been explored.

**Table 1 cells-11-02183-t001:** New therapeutic approaches targeting KRAS-Induced Cell Death Resistance.

Target/Biomarkers	Therapies	Impact on Cell Death Processes
***KRAS***^G12C^ mutation	*KRAS*^G12C^ inhibitors (ARS-853, ARS-1620, AMG-510 and MRTX849) [32,123,125,137]	Apoptosis Induction
MEK and PI3K	MEK inhibitor (AZD6244) with PI3K inhibitor (BKM120 or GDC0941) [139]	Apoptosis Induction
MEK	MEK inhibitor (trametinib) with Chemotherapy (nab-paclitaxel) [140]	Apoptosis Induction
PI3K/mTOR	Dual PI3K-mTOR inhibitor (NVP-BEZ235) [141,142]	Apoptosis Induction
RAF/ERK	pan-RAFi with an ERK-selective inhibitor [143]	Apoptosis Induction
mTOR and BCL-XL/BCL-2	mTOR inhibitor (AZD8055) with the dual BCL-XL/BCL-2 inhibitor, (ABT-263) [144]	Apoptosis Induction
MEK	Autophagy inhibitor (hydroxychloroquine) with MEK inhibitor (trametinib) [138,145]	Autophagy Inhibition
MEK	Autophagy inhibitor (hydroxychloroquine) with MEK inhibitor (binimetinib) [145,146]	Autophagy Inhibition
ERK	Autophagy inhibitor (hydroxychloroquine) with ERK inhibitor (ulixertinib) [145,146]	Autophagy Inhibition
Mutated ***KRAS***	Autophagy inhibitor (hydroxychloroquine) with KRAS inhibitor (gemcitabine) [145,146]	Autophagy Inhibition
PD-1/PD-L1	Anti-PD-1/PD-L1 monoclonal antibodies [125,133,150,151,152]	Autophagy Induction

## Data Availability

Not applicable.

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
