# Peer review of "Crucial Role of Oncogenic KRAS Mutations in Apoptosis and Autophagy Regulation: Therapeutic Implications"

_cells, 2022, doi:10.3390/cells11142183_

Round 1
Reviewer 1 Report
An outstanding review regarding the role of KRAS and its implication in apoptosis and autophagy mechanisms.
However, its implication in pancreatic ductal adenocarcinoma has not been described in this review. KRAS mutation exerts a major role in PDAC tmorigenesis and it is connected with cell death too. Please include some information regarding PDAC/KRAS/therapy
Reviewer 2 Report
Dear authors,
I read with great interest the review on the involvement of this signaling pathway in autophagy and apoptosis. I believe that this review summarizes essentials and relatively difficult informations for understanding the role of KRAS in oncogenesis and implicitly in cancer therapy.
My recommendation will be to publish with very small changes related to the form and not the substance.
As a non-specialist reader, I would like some clearer explanations of codons, the first time when they appear in the text. It may also not be uninteresting for at least one figure to show EGFR and how it activate the intracellular cascade.
I also think that some clarifications should appear at the beginning of the review regarding cross-talking between intracellular secondary pathways.
Also, in my opinion as a medical chemist, the therapeutic implications part I would like to be clearer and a little better represented.
In conclusion, I consider that the review is of great interest and I recommend publishing it with minor revisions.
Reviewer 3 Report
Comments to the authors
The article with the title “ Crucial Role of Oncogenic KRAS Mutations in Apoptosis and Autophagy Regulation: Therapeutic Implications” is in generally well done, but I would offer these comments to the investigators:
1) Some minor grammatical errors occur. The manuscript contains significant language-related issues. Please correct these types of grammatical errors throughout the paper.
2) PI3K/AKT/mTOR is a well-known autophagy inhibitory pathway. In figure 1 authors presents that this pathway is shown to activate autophagy. Please revised.
3) Line 191-192: EGFR and Autophagy. Anti-EGFR MoABs (Cetuximab/panitumumab) can activate autophagy through endocytocis as a protective mechanism. (PMID: 28323034). Please discuss it.
4) Line 316: micr-200 activates autophagy. It would be a smooth introduction to section 3. Please discuss.
5) Line 345-351: Kras and autophagy association is well know from several studies. It appears that autophagy inhibits tumor progression in early stage and helps the tumor growth in late stages where the metabolic demands are higher especially in CRC bearing mtKRAS. Please revised and discuss.
6) Line 614-625: A very interesting point for the role of autophagy in immune response for cancer cells. Autophagy regulates the expression of the neo-antigen in cancer cells. Please discuss.
7) Line 614-625: In addition with the previous comment, anti-PD-1/anti-CTLA4 and anti-EGFR antibodies appear to activate autophagy as a survival mechanism in CRC. Please discuss it (PMID: 30427914)
8) Line 610: Chloroquine and its derivative, Hydroxychloroquine have already been tested in several clinical studies. I strongly recommend to further discuss it.
9) A table with agents that activate or inhibit autophagy will help the scientific quality of the paper.
Round 2
Reviewer 3 Report
Authors addressed to all of my concerns.